# Structural basis of ligand selectivity and disease mutations in cysteinyl leukotriene receptors

Anastasiia Gusach [1,14], Aleksandra Luginina[1,14], Egor Marin[1], Rebecca L. Brouillette[2], Élie Besserer-Offroy [2], Jean-Michel Longpré[2], Andrii Ishchenko [3,4,11], Petr Popov[1,12], Nilkanth Patel[3,5], Taku Fujimoto[6], Toru Maruyama[6], Benjamin Stauch [3,4], Margarita Ergasheva[1], Daria Romanovskaia [1,13], Anastasiia Stepko [1], Kirill Kovalev[1,7,8,9,10], Mikhail Shevtsov[1], Valentin Gordeliy[1,7,8,9,10], Gye Won Han[3,4], Vsevolod Katritch [3,4,5], Valentin Borshchevskiy [1,7,9], Philippe Sarret[2]*, Alexey Mishin [1]* & Vadim Cherezov [1,3,4,5]*

Cysteinyl leukotriene G protein-coupled receptors $CysLT_1$ and $CysLT_2$ regulate pro-inflammatory responses associated with allergic disorders. While selective inhibition of $CysLT_1R$ has been used for treating asthma and associated diseases for over two decades, $CysLT_2R$ has recently started to emerge as a potential drug target against atopic asthma, brain injury and central nervous system disorders, as well as several types of cancer. Here, we describe four crystal structures of $CysLT_2R$ in complex with three dual $CysLT_1R/CysLT_2R$ antagonists. The reported structures together with the results of comprehensive mutagenesis and computer modeling studies shed light on molecular determinants of CysLTR ligand selectivity and specific effects of disease-related single nucleotide variants.

[1] Research Center for Molecular Mechanisms of Aging and Age-Related Diseases, Moscow Institute of Physics and Technology, Dolgoprudny, Russia 141701. [2] Department of Pharmacology-Physiology, Faculty of Medicine and Health Sciences, Institut de Pharmacologie de Sherbrooke, Université de Sherbrooke, Sherbrooke, QC J1H 5N4, Canada. [3] Bridge Institute, Michelson Center for Convergent Bioscience, University of Southern California, Los Angeles, CA 90089, USA. [4] Department of Chemistry, University of Southern California, Los Angeles, CA 90089, USA. [5] Department of Biological Sciences, University of Southern California, Los Angeles, CA 90089, USA. [6] Ono Pharmaceutical Co., Ltd., 3-1-1 Sakurai, Shimamoto, Mishima, Osaka 618-8585, Japan. [7] Institute of Complex Systems (ICS), ICS-6: Structural Biochemistry, Research Center Jülich, Jülich, Germany. [8] Institut de Biologie Structurale Jean-Pierre Ebel, Université Grenoble Alpes–Commissariat à l'Energie Atomique et aux Energies Alternatives–CNRS, Grenoble, France. [9] JuStruct: Jülich Center for Structural Biology, Research Center Jülich, Jülich, Germany. [10] Institute of Crystallography, RWTH Aachen University, Aachen, Germany. [11]Present address: Merck Research Laboratories, Merck & Co Inc., 770 Sumneytown Pike, West Point, PA 19486, USA. [12]Present address: Center for Computational and Data Intensive Science and Engineering, Skolkovo Institute of Science and Technology, Bolshoy Boulevard 30, Building 1, Moscow, Russia 121205. [13]Present address: CeMM Research Center for Molecular Medicine of the Austrian Academy of Sciences, 1090 Vienna, Austria. [14]These authors contributed equally: Anastasiia Gusach, Aleksandra Luginina   *email: Philippe.Sarret@USherbrooke.ca; mishinalexej@gmail.com; cherezov@usc.edu

Cysteinyl leukotrienes LTC$_4$, LTD$_4$, and LTE$_4$ are lipid mediators of inflammation acting via two G protein-coupled receptors (GPCRs), cysteinyl leukotriene receptor type 1 (CysLT$_1$R) and type 2 (CysLT$_2$R)[1]. While LTD$_4$ is the favored endogenous ligand for CysLT$_1$R[2], CysLT$_2$R responds equally to LTC$_4$ and LTD$_4$[3]. CysLTRs exhibit bronchoconstrictive and pro-inflammatory effects and, therefore, have been recognized for their role in asthma, allergic rhinitis, cardiovascular diseases, and cancers[4–7]. Several selective CysLT$_1$R antagonists, such as zafirlukast, pranlukast, and montelukast, have been approved as antiasthmatic drugs, however, a large fraction of patients does not respond to this therapy[8]. The different expression profiles, tissue distribution, and sensitivity to endogenous ligands for CysLTRs, their heterodimerization and cross regulation[9–11] as well as the prevalence of asthma-associated polymorphisms in CysLT$_2$R[12,13] suggest distinct roles for each receptor subtype in physiology and pathology. Based on an LTC$_4$-induced animal asthma model, it was proposed that CysLT$_2$R-selective or dual antagonists may improve treatments of severe asthma cases[14]. Furthermore, selective inhibition of CysLT$_2$R predominantly expressed in cardiovascular and brain tissues has shown remedial effects in ischemic conditions and acute brain injuries[15]. The development of more efficient therapies against asthma and related diseases is hampered by the lack of specific knowledge about selectivity and functional mechanisms of CysLTRs, which requires high-resolution structural data. Here, we describe four crystal structures of CysLT$_2$R in complex with three dual CysLT$_1$R/CysLT$_2$R antagonists (cpds 11a–c, Supplementary Fig. 1, Supplementary Methods) and the results of extensive mutagenesis and computer modeling studies. Along with recently reported structures of CysLT$_1$R in complex with zafirlukast and pranlukast[16], we now have a complete structural view of receptors mediating action of cysteinyl leukotrienes in their inhibited, inactive state.

## Results

**CysLT$_2$R structure determination.** To facilitate crystallization, human CysLT$_2$R was modified by truncating N- and C-termini, inserting a thermostabilized apocytochrome b$_{562}$RIL[17] into the intracellular loop 3 (ICL3), and introducing three stabilizing point mutations[18]: W51$^{1.45}$V, D84$^{2.50}$N, and F137$^{3.51}$Y (superscript refers to the Ballesteros–Weinstein GPCR residue numbering scheme[19]). The engineered receptor was crystallized in lipidic cubic phase (LCP)[20] in complex with three antagonists: ONO-2570366 (cpd 11a) (2.4 Å resolution in two different space groups), ONO-2770372 (cpd 11b; 2.7 Å), and ONO-2080365 (cpd 11c; 2.7 Å) (Supplementary Figs. 1–4 and Supplementary Table 1). To validate the structures and probe the role of key residues, involved in ligand binding and receptor function, we conducted cell surface expression and IP$_1$ stimulation and inhibition assays with a set of 24 mutants (Supplementary Figs. 5 and 6 and Table 1).

**Overall architecture of CysLT$_2$R.** All CysLT$_2$R structures adopt the canonical seven-transmembrane helical bundle architecture (Fig. 1a) and are structurally similar to CysLT$_1$R-pranlukast[16] (Supplementary Table 2). Overall CysLT$_2$R conformations are identical to each other (Supplementary Table 2), except for the structure with cpd 11c, which is described below. Our further analysis, therefore, is focused on the highest resolution CysLT$_2$R-11a structure, unless noted otherwise. Extracellular loop 2 (ECL2) in CysLT$_2$R is stabilized by the highly conserved disulfide bond[21] between C111$^{3.25}$ and C187$^{ECL2}$. An additional disulfide bond is formed between C31$^{1.25}$ and C279$^{7.27}$ (Fig. 1c). Notably, both

TM1 and TM7 are about one helical turn shorter than in CysLT$_1$R, resulting in a ~5 Å shift of ECL3 tip (Fig. 1b).

As expected, CysLT$_2$R structures with antagonists 11a and 11b are captured in a fully inactive state. Similar to inactive structures of CysLT$_1$R and other receptors from the δ-branch of class A GPCRs, the P$^{5.50}$-I$^{3.40}$-F$^{6.44}$ microswitch is found in a distinct conformation (Fig. 1e)[16], previously associated with activation of receptors from other class A GPCR branches. The role of this microswitch in receptors from the δ-branch is apparently different and is likely linked to the substitution of the "toggle switch" W$^{6.48}$ with F$^{6.48}$, which prevents this microswitch from accessing its inactive conformation. The highly conserved D[E]R$^{3.50}$Y motif, in which R$^{3.50}$ is stabilized in an inactive conformation via a salt bridge with D[E]$^{3.49}$, is replaced by VR$^{3.50}$F in CysLT$_2$R (Fig. 1f). As expected, restoring the canonical ionic lock by V135$^{3.49}$D in CysLT$_2$R decreases the potency of LTD$_4$ while increasing the potency of antagonists through stabilization of the inactive conformation (Table 1). Restoration of Y in the D[E]RY motif via F137$^{3.51}$Y mutation, which is also present in the crystallized construct, has no effect on the potency of LTD$_4$ or antagonists. Similarly, the stabilizing mutation W51$^{1.45}$V in the crystallized construct has little effect on ligand binding and receptor signaling. Finally, the third crystallization construct mutation D84$^{2.50}$N, a known stabilizing mutation in the conserved in class A GPCRs sodium-binding pocket[22–24], abolishes LTD$_4$-stimulated IP$_1$ production in CysLT$_2$R, similar to its effect in other receptors[25]. Likewise, N297$^{7.45}$C in the sodium-binding pocket results in a complete loss of signaling activity. Mutating N301$^{7.49}$D in the conserved NP$^{7.50}$xxY motif (Fig. 1d) (NPLLY in CysLT$_2$R; DPLLY in CysLT$_1$R) stabilizes the sodium-binding pocket and thus reduces LTD$_4$ signaling potency 6-fold, while increasing receptor surface expression and $E_{max}$ (Table 1).

Interestingly, the CysLT$_2$R-11c structure shows a different orientation of the Y221$^{5.58}$ microswitch along with a distinct conformation of the intracellular part of TM6, shifted ~5 Å outward compared with other CysLT$_2$R structures (Supplementary Fig. 7a). Both changes are consistent with a partially active-like GPCR state[26], which, however, lacks key activation-related changes in TM7 and sodium pocket. Molecular dynamics (MD) simulations show that this state is distinct from both active and inactive states and highly dynamic (Supplementary Fig. 7b, c), suggesting that CysLT$_2$R-11c likely represents an intermediate conformational state, selected and stabilized by the crystal lattice.

Unlike CysLT$_1$R structures, all CysLT$_2$R structures, except for the complex with cpd 11c, possess a well-resolved intracellular amphipathic helix 8 (H8) running parallel to the membrane (Fig. 1b). While the function of H8 is not fully understood, a mounting evidence points toward its importance in the regulation of G protein and β-arrestin binding[27,28]. Notably, the junction between TM7 and H8 in CysLT$_1$R contains a rare GG$^{8.48}$ motif, which likely increases dynamics of H8. On the other hand, position 8.48 in CysLT$_2$R is occupied by E310$^{8.48}$, which stabilizes the junction and the inactive state by forming salt bridges with R136$^{3.50}$ and K244$^{6.32}$ (Fig. 1f). Removing these interactions by E310$^{8.48}$A or E310$^{8.48}$G results in a slightly increased potency of LTD$_4$ in IP$_1$ signaling assays (Table 1).

**Ligand-binding pocket and ligand-receptor interactions.** In all CysLT$_2$R structures, a strong electron density for the ligand (Supplementary Fig. 4) is present inside the central cavity of the receptor that consists of residues from all seven TMs and ECL2. It has a narrow opening (~3 Å diameter) between ECLs into the extracellular space and a larger access cleft (~5 Å across) from the lipid bilayer between TM4 and TM5 (Fig. 2a). All antagonists

**Table 1 Signaling and cell surface expression data for CysLT$_2$R.**

| Mutation | Mutation location | LTD$_4$ EC$_{50}$ ± s.d. nM | LTD$_4$ $E_{max}$ ± s.d. % of WT | Cpd 11a IC$_{50}$ ± s.d. nM | Cpd 11c IC$_{50}$ ± s.d. nM | Cell surface expression % of WT ± s.d. |
|---|---|---|---|---|---|---|
| Wild type | | 4.4 ± 0.8 [4] | 100 ± 7 [4] | 14 ± 6 [5] | 70 ± 20 [4] | 100 ± 50 [5] |
| K37$^{1.31}$R | Ligand-binding pocket | 40 ± 20 [2] | 110 ± 20 [2] | 80 ± 30 [2] | 280 ± 100 [2] | 43 ± 8 [2] |
| W51$^{1.45}$V | CC | 6 ± 5 [2] | 180 ± 60 [2] | 47 ± 10 [2] | 160 ± 30 [2] | 540 ± 100 [2] |
| D84$^{2.50}$N | CC, sodium pocket | N/R [2] | N/R [2] | ND | ND | 480 ± 190 [2] |
| Y119$^{3.33}$F | Ligand-binding pocket | 70 ± 30 [2] | 90 ± 30 [2] | 100 ± 40 [2] | 260 ± 100 [2] | 71 ± 10 [2] |
| I126$^{3.40}$V | PIF motif | 6 ± 4 [2] | 130 ± 70 [2] | 20 ± 10 [2] | 14 ± 5 [3] | 63 ± 8 [2] |
| Y127$^{3.41}$W | Ligand-binding pocket | 1.2 ± 0.4 [2] | 110 ± 30 [2] | 500 ± 400 [2] | 44 ± 10 [2] | 240 ± 50 [2] |
| L129$^{3.43}$Q | Disease related, sodium pocket | N/R [2] | N/R [2] | ND | ND | 98 ± 14 [2] |
| V135$^{3.49}$D | DRY motif | 20 ± 10 [2] | 50 ± 20 [2] | 3 ± 3 [2] | 40 ± 30 [2] | 35 ± 15 [2] |
| F137$^{3.51}$Y | CC, DRY motif | 8 ± 2 [2] | 120 ± 10 [2] | 15 ± 5 [2] | 70 ± 20 [2] | 150 ± 20 [2] |
| S169$^{4.56}$A | Ligand-binding pocket | 5 ± 1 [2] | 120 ± 30 [2] | 10 ± 4 [3] | 16 ± 10 [3] | 130 ± 50 [2] |
| K194$^{ECL2}$R | Ligand-binding pocket | 50 ± 20 [2] | 140 ± 30 [2] | 60 ± 30 [2] | 40 ± 20 [2] | 150 ± 40 [2] |
| K194$^{ECL2}$N | Ligand-binding pocket | 5 ± 2 [2] | 110 ± 20 [2] | 8 ± 4 [2] | 32 ± 10 [2] | 47 ± 13 [2] |
| L198$^{5.35}$A | Ligand-binding pocket | N/R [2] | N/R [2] | ND | ND | 150 ± 30 [2] |
| M201$^{5.38}$A | Ligand-binding pocket | N/R [2] | N/R [2] | ND | ND | 150 ± 80 [2] |
| M201$^{5.38}$L | Ligand-binding pocket | N/R [2] | N/R [2] | ND | ND | 90 ± 30 [2] |
| M201$^{5.38}$V | Disease related, ligand-binding pocket | 100 ± 50 [2] | 70 ± 10 [2] | 30 ± 10 [3] | 60 ± 20 [3] | 80 ± 40 [2] |
| N202$^{5.39}$H | Ligand-binding pocket | >1000 [2] | 50 ± 20 [2] | 290 ± 100 [3] | 60 ± 30 [3] | 110 ± 20 [2] |
| F260$^{6.48}$W | Toggle switch | 7 ± 5 [2] | 50 ± 30 [2] | 40 ± 20 [2] | 330 ± 100 [2] | 38 ± 7 [2] |
| R267$^{6.55}$K | Ligand-binding pocket | 90 ± 60 [2] | 50 ± 20 [2] | 22 ± 9 [3] | 100 ± 30 [3] | 33 ± 15 [2] |
| H284$^{7.32}$Q | Ligand-binding pocket | 21 ± 6 [2] | 120 ± 10 [2] | 230 ± 90 [2] | 270 ± 200 [2] | 170 ± 20 [2] |
| N297$^{7.45}$C | Sodium pocket | N/R [2] | N/R [2] | ND | ND | 105 ± 11 [2] |
| N301$^{7.49}$D | Sodium pocket | 25 ± 7 [2] | 120 ± 40 [2] | 170 ± 70 [2] | 110 ± 50 [2] | 180 ± 40 [2] |
| E310$^{8.48}$A | Helix 8 | 1.6 ± 0.7 [2] | 60 ± 20 [2] | 8 ± 3 [2] | 18 ± 5 [2] | 10 ± 9 [2] |
| E310$^{8.48}$G | Helix 8 | 0.9 ± 0.3 [2] | 90 ± 20 [2] | 2 ± 1 [2] | 7 ± 6 [2] | 26 ± 4 [2] |
| 3-Mut | CC | N/R [2] | N/R [2] | ND | ND | 190 ± 120 [2] |
| 3-Mut-ΔC | CC | N/R [2] | N/R [2] | ND | ND | 110 ± 30 [2] |
| CC | CC | N/R [2] | N/R [2] | ND | ND | ≥1,000 [2] |

Data are shown for LTD$_4$-induced IP$_1$ accumulation (agonist potency, EC$_{50}$, and efficacy, $E_{max}$) and for inhibition of LTD$_4$-induced IP$_1$ production by antagonists (IC$_{50}$ values). Cell surface expression is determined by ELISA using anti-HA tag antibody, normalized to the expression of WT CysLT$_2$R. All data are shown as mean ± s.d. (number of independent experiments performed in quadruplicates in brackets). All nonresponsive to LTD$_4$ stimulation mutants (N/R) are expressed at the cell surface. ND—not determined, because these mutants did not respond to LTD$_4$ stimulation. "3-Mut"—three mutations W51$^{1.45}$V, D84$^{2.50}$N, and F137$^{3.51}$Y used in the crystallized construct (CC). "3-Mut-ΔC"—three mutations W51$^{1.45}$V, D84$^{2.50}$N, and F137$^{3.51}$Y and Δ323–346 C-terminal truncation, as in CC

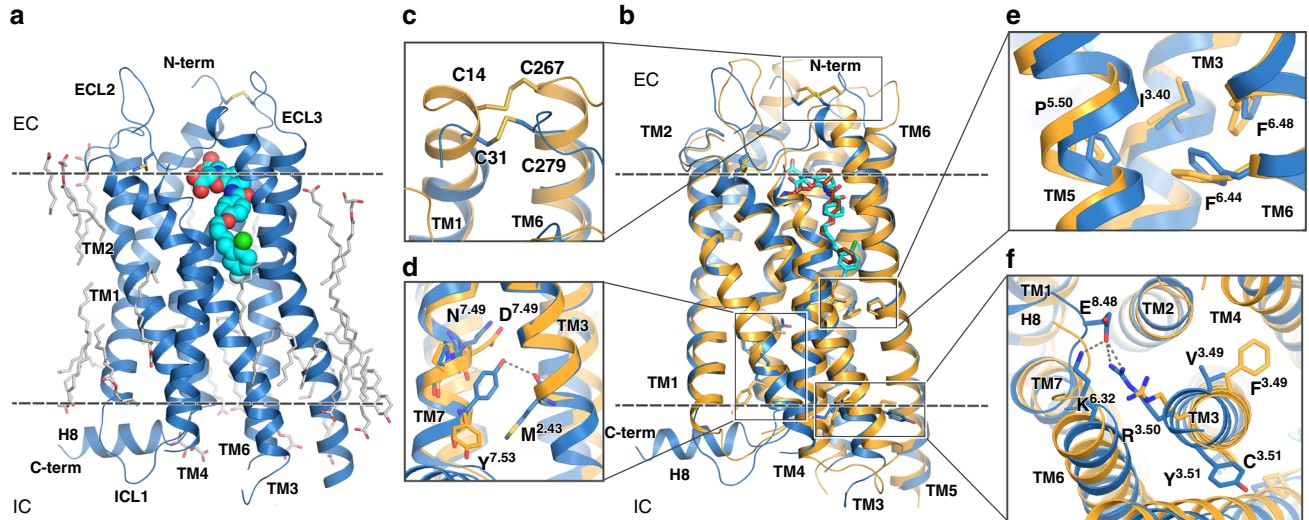

**Fig. 1 Structure of CysLT$_2$R. a** Overall structure of CysLT$_2$R-11a (C222$_1$ space group). **b** Structural superposition of CysLT$_2$R-11a (blue; C222$_1$ space group) with CysLT$_1$R-pranlukast (yellow). **c** Comparison of disulfide bridges between CysLT$_1$R (yellow) and CysLT$_2$R (blue). Comparison of functional motifs: NPxxY (**d**) P-I-F (**e**), and DRY (**f**). Membrane boundaries are shown as dashed lines in **a** and **b**.

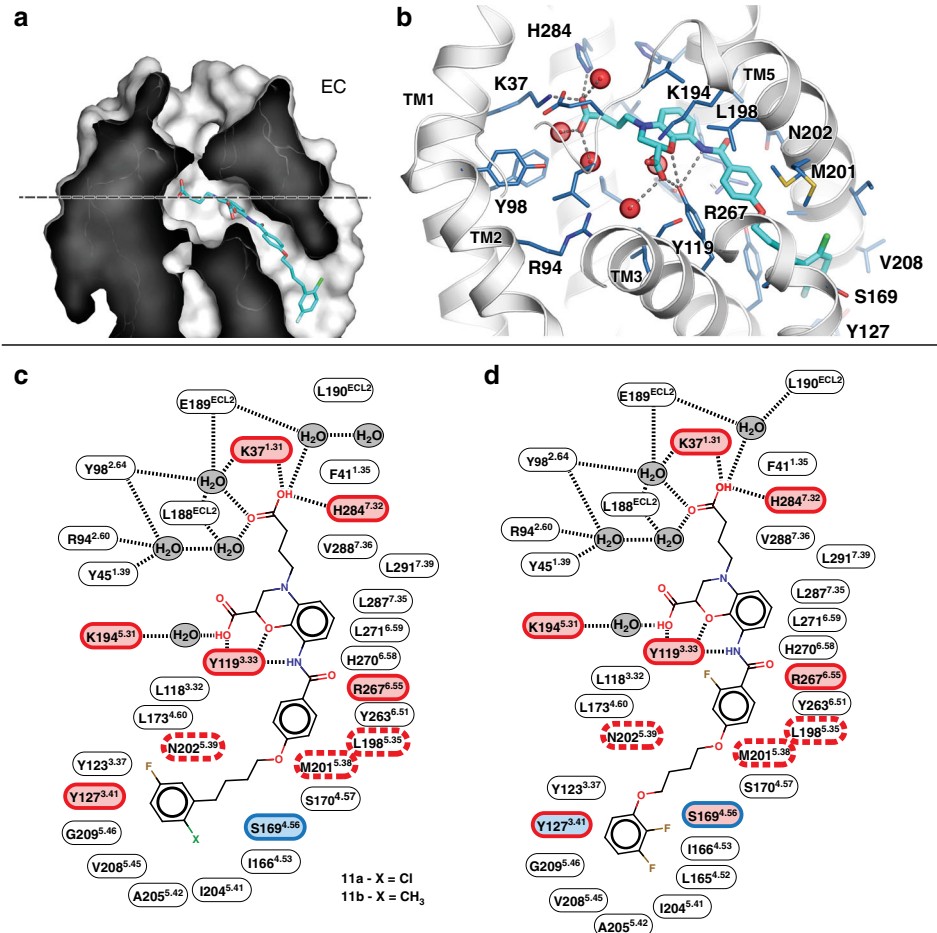

**Fig. 2 Ligand-binding pocket of CysLT$_2$R. a** Sliced surface representation of the ligand-binding pocket in CysLT$_2$R. **b** Binding pose of cpd 11a and details of ligand-receptor interactions. Schematic diagrams of CysLT$_2$R interactions with cpds 11a and 11b (**c**) and cpd 11c (**d**). Residues are colored according to the effect of their mutations on the antagonist potency in IP$_1$ signaling assays: light red—strong effect, blue—no effect, white—not tested. The outline color indicates the effect of mutations on LTD$_4$ potency: red—strong effect, red dashed—nonresponsive mutants, blue—no effect.

cocrystallized with CysLT$_2$R share the same 3,4-dihydro-2H-1,4-benzoxazine scaffold and bind in the pocket in similar conformations (root mean square deviation < 0.3 Å in the common scaffold, Fig. 2). A key anchoring residue Y119[3.33], conserved in CysLTRs, forms multiple polar contacts with the benzoxazine part, carboxylic group, and amide linker of all ligands (Fig. 2b–d). Y119[3.33]F mutant shows decreased potencies for both LTD$_4$ and antagonists in IP$_1$ assay (Table 1). The N-linked carboxypropyl moiety makes salt bridges with K37[1.31] and H284[7.32] that are specific to CysLT$_2$R. Mutating these residues to their CysLT$_1$R counterparts (K37[1.31]R or H284[7.32]Q) drastically decreases potencies for LTD$_4$ activation as well as inhibition by antagonists, suggesting distinct binding interactions of these ligands with CysLT$_1$R and CysLT$_2$R.

The hydrophobic bottom part of the ligand-binding cleft containing the butoxybenzene group of cpd 11a is formed by side chains of TM3-TM5 and, in case of cpd 11c, extends to L165[4.52] and I166[4.53]. Y127[3.41] forms an interhelical hydrogen bond with the carbonyl oxygen of Val208[5.45], stabilizing a Pro-induced kink in TM5 (Fig. 2c, d). An aromatic residue in position 3.41 at the intersection of TM3-TM5 was previously described to confer receptor stabilization[29]. Interestingly, mutation Y127[3.41]W slightly improves CysLT$_2$R surface expression and potencies of LTD$_4$ and cpd 11c, however, dramatically decreases the potency of cpd 11a to inhibit LTD$_4$-induced IP$_1$ accumulation, likely because of a clash between bulky tryptophan and 2-chloro-5-fluoro-phenyl group of cpd 11a. S169[4.56] forms a hydrogen bond

with the carbonyl group of L165[4.52] and interacts with the fluorine atom of cpd 11c phenyl group. Mutation S169[4.56]A does not affect EC$_{50}$ for LTD$_4$ and IC$_{50}$ for cpd 11a but moderately improves inhibition by cpd 11c.

**Antagonist selectivity to CysLTR subtypes**. To understand the mechanism of ligand selectivity, we performed docking of 18 derivatives of the common 3,4-dihydro-2H-1,4-benzoxazine-2-carboxylic acid scaffold[30,31] with a large spectrum of CysLT$_1$R/CysLT$_2$R selectivity (Supplementary Table 3). Docking models of the most selective compounds in this structure-activity relationship (SAR) series[30], cpd 13e (1,800-fold selective for CysLT$_1$R) and cpd 15b (200-fold selective for CysLT$_2$R), are shown in Fig. 3, alongside with cpd 11a (dual CysLT$_1$R/CysLT$_2$R), cocrystallized with CysLT$_2$R, and pranlukast (4,500-fold selective for CysLT$_1$R as shown in Supplementary Fig. 6a), cocrystallized with CysLT$_1$R.

SAR analysis revealed that the most important factor for CysLT$_2$R selectivity is the length of the alkyl chain for the O-substituents (R$^1$), where longer phenylpentyl group in cpd 15b achieves much higher CysLT$_2$R selectivity than phenylbutyl in cpd 11a, cpd 13e, and pranlukast or phenylpropyl in some other compounds such as cpd 15a (Supplementary Table 3). Comparison of the contacts of these substituents in CysLT$_1$R-pranlukast and CysLT$_2$R-11a suggests that in CysLT$_1$R the cleft opening to the lipid membrane is restricted by a hydrophobic ridge formed by F150[4.52], F112[3.41], and V196[5.45], while in CysLT$_2$R the

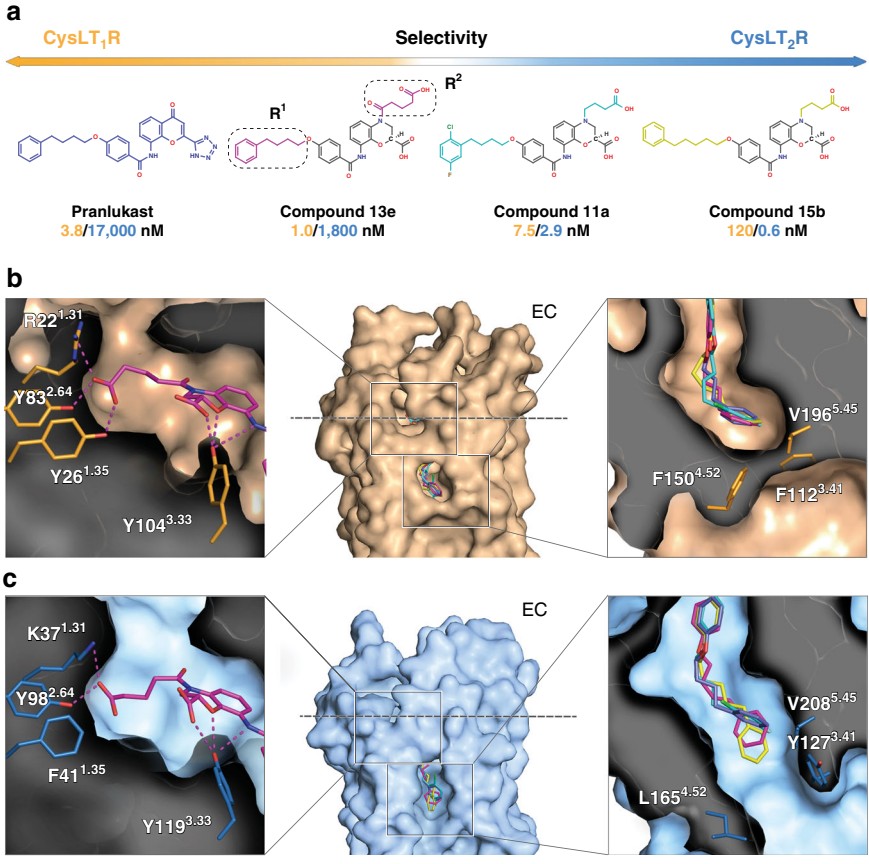

**Fig. 3 Structural determinants of antagonist selectivity to CysLTR subtypes. a** Examples of compounds used in the docking studies, with their IC₅₀ values toward CysLT₁R and CysLT₂R shown in yellow and blue, respectively. IC₅₀ values for pranlukast were obtained in this work (3.8 ± 0.7 nM (CysLT₁R) and ~17,000 ± 12,000 nM (CysLT₂R), expressed as mean ± s.d. of three independent experiments, tested in quadruplicate) and for other ligands were quoted from ref. [30]. The common 3,4-dihydro-2H-1,4-benzoxazine-2-carboxylic acid scaffold is shown in gray. Overview of the ligand-binding pocket with the docked ligands for CysLT₁R (**b**) and CysLT₂R (**c**). Inserts show docking poses and details of ligand interactions with CysLT₁R and CysLT₂R.

replacement F$^{4.52}$L removes this restriction, making the cleft more open. Accordingly, docking of cpd 15b into CysLT₁R results in a strained alkyl chain and a clash of the terminal phenyl group with F150$^{4.52}$, while in CysLT₂R the phenyl group readily extends outside of the cleft (Fig. 3b, c). Moreover, the phenylbutyl group in this and other scaffolds tolerates methyl and halogen decorations in the ortho and meta positions, which enables tuning pharmacological properties of the ligand such as solubility and stability, as exemplified by the development of gemilukast[32].

SAR analysis of the N-substituent (R$^2$) suggests that its length as well as the presence of a carboxyl group in this scaffold has critical influence on IC₅₀ values for both CysLTRs. Indeed, docking of cpd 13e, the most selective antagonist in this series, shows that the oxo-pentanoic-acid moiety of this ligand forms a hydrogen bond with Y26$^{1.35}$, while CysLT₂R has F41$^{1.35}$ at this position and cannot form a hydrogen bond with the ligand (Fig. 3b, c). Further elongation of this derivative chain is limited by the size of this subpocket. Interestingly, removal of the carbonyl group, as in cpds 14a-c and 15b, shifts selectivity toward CysLT₂R, suggesting that a flexible carboxy-alkyl chain is favored for this receptor[30]. Altogether, CysLT₁R and CysLT₂R crystal structures provide atomic level insights into the mechanisms of ligand recognition and subtype selectivity. This knowledge should contribute to the rational design of more efficient antagonists with improved affinity/efficacy or subtype selectivity profiles.

**Structural insights into CysLT₂R disease-related mutations.** Finally, our structures provide rational explanations of the two

most common disease-associated single-nucleotide variants (SNVs) in CysLT₂R: M201$^{5.38}$V, related to atopic asthma[13,33], and the oncogenic L129$^{3.43}$Q mutation[34,35]. M201$^{5.38}$ together with M172$^{4.59}$, L173$^{4.60}$, and L198$^{5.35}$ define the shape of the hydrophobic part of the ligand-binding pocket. Substitutions of L198$^{5.35}$ with alanine or M201$^{5.38}$ with alanine or leucine result in nonresponsive mutants that bind LTD₄ but fail to stimulate IP₁ production. In contrast to the alanine or leucine substitution, the atopic asthma-associated variant M201$^{5.38}$V still responds to LTD₄ stimulation. However, this mutation significantly decreases LTD₄ potency and efficacy to induce IP₁ accumulation when compared with the wild-type CysLT₂R (Table 1). These results along with a similar effect of N202$^{5.39}$H suggest the importance of ligand-dependent TM5 displacement in CysLT₂R activation. Indeed, all three TM5 residues (L198$^{5.35}$, M201$^{5.38}$, and N202$^{5.39}$) that are important for potency interact with the benzamide core of antagonists, which distinguish them from agonists, and thus likely modulate TM5 conformation and dynamics that control activation (Fig. 4a).

The second disease-relevant SNV, L129$^{3.43}$Q, has been associated with uveal melanoma and blue nevi[34–36]. A hydrophobic amino acid is present in this position in 97% of class A receptors, most frequently L$^{3.43}$ (73%), but also M$^{3.43}$ as in CysLT₁R. Located at the bottom of the sodium pocket, a large hydrophobic side chain in position 3.43 is part of a hydrophobic layer, which is important for stability of the inactive state. Mutation of L$^{3.43}$ to a polar residue or to a small alanine residue can disrupt the hydrophobic layer (Fig. 4b, c), facilitating water and sodium passage[37,38] and leading

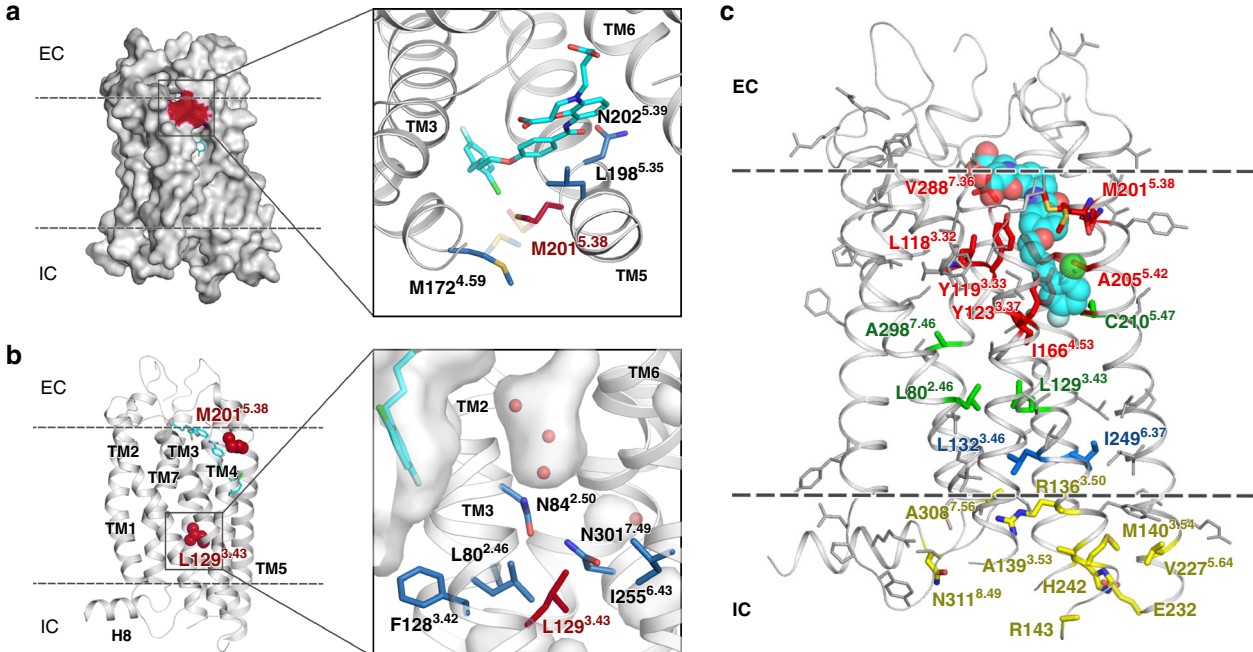

**Fig. 4 Naturally occurring missense SNVs, mapped on the CysLT$_2$R structure. a** M201$^{5.38}$V polymorphism, associated with atopic asthma. **b** L129$^{3.43}$Q mutation, related to uveal melanoma and blue nevi. **c** SNVs from the ExAC database and L129$^{3.43}$, colored according to their location: ligand-binding pocket (red), microswitches (blue), sodium site (green), and G protein and β-arrestin-binding interface (yellow).

to receptor activation. Indeed, it was shown that mutation in position 3.43 to R, K, A, E, or Q induces constitutive activation in several receptors[39], often resulting in distinct physiological disorders. In CysLT$_2$R, we found that L129$^{3.43}$Q displays constitutive activity for the G$_q$ pathway with a fourfold increase in basal IP$_1$ accumulation and is unresponsive to LTD$_4$ stimulation (Supplementary Fig. 6b, c).

Further, we evaluated naturally occurring missense SNVs in CysLT$_2$R from over 60,000 healthy individuals assembled in the exome aggregation consortium (ExAC) database[40]. Structural mapping of 117 SNV positions revealed that nine of them belong to the ligand-binding pocket, seven are activation-related microswitches or located in the sodium-binding site, and nine reside on the G protein and β-arrestin-binding interface (Fig. 4c), all of which could dramatically affect the receptor function[41]. Unlike the relatively frequent polymorphisms M201$^{5.38}$V or L129$^{3.43}$Q, most ExAC mutations are very rare (minor allele frequency < 10$^{-4}$), and, therefore, it has not been possible yet to associate them with higher risk of asthma or other pathologies.

## Discussion

Compared with CysLT$_1$R, which was successfully targeted by antiasthmatic drugs 20 years ago, the role of CysLT$_2$R in physiology and pathogenesis of inflammation related processes is more complex and remains less understood[4,42]. Recently accumulated results suggest that CysLT$_2$R-selective or CysLT$_1$R/CysLT$_2$R dual antagonists may offer more efficient alternatives to currently used CysLT$_1$R-selective antagonists, especially for the treatment of severe asthma[43,44]. In addition, CysLT$_2$R is arising as a promising drug target against brain injury and neurodegenerative disorders[5,45]. High constitutive G$_q$ signaling activity of CysLT$_2$R mutants has been associated with occurrence of uveal melanoma and other cancer types[35], however, the role of CysLT$_2$R in cancer remains controversial as its high expression levels have been correlated with antitumorigenic activity[46].

The CysLT$_2$R structures described in this study along with the structures of CysLT$_1$R[16] reveal important determinants of ligand binding and selectivity between these two receptors. Thus, our docking studies recapitulate binding of dozens of known ligands and allow to explain SAR for a series of 3,4-dihydro-2H-1,4-benzoxazine-2-carboxylic acid scaffold derivatives. These structures will serve as templates for rational design of a new generation of potent antagonists with desired selectivity profiles (receptor selective or dual), which could be further developed into efficient drug candidates or tool compounds, helping to decipher the specific role of each of the CysLT receptor subtype in various physiological processes and pathologies.

Our study also provides a key insight into structure and function of the intracellular H8 in CysLT receptors. While both receptors possess a canonical H8 amphipathic motif, this helix is well resolved in CysLT$_2$R structure, but not observed in CysLT$_1$R (Fig. 1). The difference is that the junction between TM7 and H8 in CysLT$_1$R contains a very flexible GG$^{8.48}$ motif, while CysLT$_2$R has GE$^{8.48}$ in the same position. Importantly, G$^{8.48}$S mutation in CysLT$_1$R is a known disease mutation that increases efficacy of the receptor signaling[16,47], likely due to the improved stability of H8, known to be involved in regulation of G protein and β-arrestin binding[27,28]. Interestingly, the E310$^{8.48}$ side chain in CysLT$_2$R serves a special role, forming salt bridges with R136$^{3.50}$ and K244$^{6.32}$ and thus stabilizing the inactive receptor state. Introduction of Glu in positions 8.48 and 8.49 has been recently shown beneficial effects on stability of the inactive state in several GPCRs, including CB2[48] and CCR5[49].

Another promising application for structural information obtained in this study is the ability to rationalize effects of specific SNVs on receptor function. We mapped naturally occurring missense SNVs from 60,000 healthy individuals on the CysLT$_2$R structure and observed that about quarter of them are located in functionally important regions, which may affect signaling[40]. Continuing increase in structural coverage of the GPCR superfamily combined with rapid accumulation of genome sequencing data and

structure-function studies should enable reliable predictions of disease associations and effects of natural missense variants on drug efficacy and safety profiles, advancing us toward the realm of personalized medicine.

## Methods

**Protein engineering for structural studies.** The wild-type DNA encoding human cysteinyl leukotriene receptor 2 (UniProt Q9NS75) was purchased from cDNA Resource Center (cdna.org) and cloned into a modified pFastBac1 vector (Invitrogen) containing an expression cassette with a haemagglutinin signal sequence, FLAG tag, $10 \times$ His tag followed by TEV protease cleavage site on the N-terminus. Amino acids 1–16 from the N-terminus and 323–346 from C-terminus were deleted by overlap extension PCR. Thermostabilized apocytochrome $b_{562}$RIL (BRIL) from *Escherichia coli* with mutations M7W, H102I, and R106L was inserted into the ICL3 between the residues E232 and V240 by overlap extension PCR. Three point mutations, $W51^{1.45}V$, $D84^{2.50}N$, and $F137^{3.51}Y$, designed using a sequence dissimilarity approach[18], were further introduced to improve receptor surface expression in *Spodoptera frugiperda* Sf9 cells (Novagen, cat. 71104) as well as its stability and yield. Sequences of all primers used in this work are listed in Supplementary Table 4. The full DNA sequence of the CysLT$_2$R crystallization construct is provided in Supplementary Table 5.

**Protein expression and purification.** Bac-to-Bac baculovirus expression system (Invitrogen) was used to obtain high-titer recombinant baculovirus ($>3 \times 10^8$ viral particles per ml). *Sf9* insect cells were infected at densities ($2$–$3) \times 10^6$ cells per ml culture at multiplicity of infection of 5–10. BayCysLT2 ligand (Cayman Chemical) was dissolved in DMSO to 25 mM and added to the cell culture at the time of infection. Cells were harvested 48–50 h post infection by gentle centrifugation at $2,000 \times g$ and stored at $-80\,°C$ until use.

Cells were thawed and lysed by repetitive washes in hypotonic buffer (10 mM HEPES pH 7.5, 20 mM KCl, and 10 mM $MgCl_2$) and high osmotic buffer (10 mM HEPES pH 7.5, 20 mM KCl, 10 mM $MgCl_2$, and 1 M NaCl) with addition of protease inhibitor cocktail (500 μM 4-(2-aminoethyl)benzenesulfonyl fluoride hydrochloride (Gold Biotechnology), 1 μM E-64 (Cayman Chemical), 1 μM leupeptin (Cayman Chemical), 150 nM aprotinin (A.G. Scientific)). Membranes were then resuspended in 10 mM HEPES pH 7.5, 20 mM KCl, 10 mM $MgCl_2$, 2 mg ml$^{-1}$ iodoacetamide, protease inhibitors, and 25 μM ligand for 30 min at 4 °C and then solubilized by addition of 2× buffer (300 mM NaCl, 2% of n-dodecyl-β-D-maltopyranoside (DDM; Avanti Polar Lipids) 0.4% of cholesteryl hemisuccinate (CHS; Sigma), 10% glycerol) and incubation for 3.5 h at 4 °C. All further purification steps were performed at 4 °C. Supernatant was clarified by centrifugation and bound to TALON IMAC resin (Clontech) overnight in presence of 20 mM imidazole and NaCl added up to 800 mM. The resin was then washed with ten column volumes (CV) of wash buffer I (8 mM ATP, 100 mM HEPES pH 7.5, 10 mM $MgCl_2$, 500 mM NaCl, 15 mM imidazole, 10 μM ligand, 10% glycerol, 0.1/0.02% DDM/CHS), then with five CV of wash buffer II (25 mM HEPES pH 7.5, 500 mM NaCl, 30 mM imidazole, 10 μM ligand, 10% glycerol, 0.015/0.003% DDM/CHS), then buffer was exchanged into buffer III (25 mM HEPES pH 7.5, 500 mM NaCl, 10 mM imidazole, 10 μM ligand, 10% glycerol, 0.05/0.01% DDM/CHS) and the protein-containing resin was treated with PNGase F (Sigma) for 5 h. Resin was further washed with five CV of wash buffer III and eluted with (25 mM HEPES pH 7.5, 250 mM NaCl, 400 mM imidazole, 10 μM ligand, 10% glycerol, 0.05/0.01% DDM/CHS) in several fractions. Fractions containing target protein were desalted from imidazole using PD10 desalting column (GE Healthcare) and incubated with 50 μM ligand and a His-tagged TEV protease (homemade) overnight to remove the N-terminal tags. Reverse IMAC was performed the following day and protein was concentrated up to 40–60 mg ml$^{-1}$ using a 100 kDa molecular weight cut-off concentrator (Millipore). The protein purity was checked by SDS-PAGE, and the protein yield and monodispersity were estimated by analytical size exclusion chromatography.

**LCP crystallization.** Purified and concentrated CysLT$_2$R was reconstituted in LCP, made of monoolein (Nu-Chek Prep) supplemented with 10% (w/w) cholesterol (Affymetrix) in 2:3 protein:lipid ratio using a lipid syringe mixer[20]. Transparent LCP mixture was dispensed onto 96-wells glass sandwich plates (Marienfeld) in 25–40 nl drops and covered with 800 nl precipitant using an NT8-LCP robot (Formulatrix). All LCP manipulations were performed at room temperature (RT, 20–23 °C), and plates were incubated and imaged at 22 °C using an automated incubator/imager (RockImager 1000, Formulatrix). Crystals of CysLT$_2$R-11a_C222$_1$ grew to their full size within 3 weeks in a precipitant containing 100–200 mM NH$_4$ tartrate dibasic, 28–32% v/v PEG400, and 100 mM HEPES pH 8.0; CysLT$_2$R-11a_F222 for 3 weeks in a precipitant containing 30 mM NH$_4$ tartrate dibasic, 24% PEG400, and 100 mM HEPES 7.0; CysLT$_2$R-11b for 3 weeks in a precipitant containing 210 mM NH$_4$ tartrate dibasic, 29% PEG400, and 100 mM HEPES 7.0; and CysLT$_2$R-11c for 1 week in a precipitant containing 100 mM K formate, 30% v/v PEG400, and 100 mM TRIS-HCl pH 8.0. Crystals were harvested from LCP using 75–200 μm MiTeGen micromounts and flash-frozen in liquid nitrogen.

**Diffraction data collection and structure determination.** X-ray diffraction data were collected at the European Synchrotron Radiation Facility (ESRF, Grenoble, France) beamlines ID23–1, ID29, ID30b, and ID30a3, equipped with PILATUS3 6M, PILATUS3 6M-F, or Eiger X 4M detectors, using the X-ray wavelengths in range 0.96770–1.07234 Å and the beam size between 15 and 30 μm. In case of CysLT$_2$R-11c, four partial (70–80°) datasets with oscillation 0.2° and three partial 20° datasets with oscillation 0.1° were collected and combined to obtain a complete final dataset. The exposure was calculated using RADDOSE[50] based on a dose of 20 MGy per dataset, as implemented in BEST[51]. For CysLT$_2$R-11a and CysLT$_2$R-11b, partial datasets of 5–15° per crystal with oscillations of 0.1–0.15° per image and the exposure time set to reach 20 MGy dose for each partial dataset were collected following a raster scanning of each crystal and selection of best diffraction spots using DOZOR scoring[52] and manual inspection of diffraction images. Data were integrated using XDS, scaled and merged with XSCALE[53], nonisomorphous datasets were rejected using $CC_{1/2}$-based clustering as previously described[52]. The structure was determined by molecular replacement using phenix.phaser[54] with the receptor portion of CysLT$_1$R-pran (PDB ID 6RZ4) and BRIL of $A_{2A}$AR (PDB ID 4EIY) as models for the initial cpd 11a structure, and this model was subsequently used as the molecular replacement search model for the three other structures. Initial refinement rounds were performed using autoBUSTER[55] and at later stages with phenix.refine[56], followed with manual examination and rebuilding with COOT[54] using both 2mFo-DFc and mFo-DFc maps. Final data collection and refinement statistics are shown in Supplementary Table 1.

**Plasmids for functional assays.** For CysLT$_2$R functional assays, the initial CysLT$_2$R wild-type gene with an N-terminal $3 \times$HA tag cloned into pcDNA3.1+ (Invitrogen) at EcoRI(5′) and XhoI(3′) was purchased from cDNA.org. All further gene modifications (point mutations, truncations, or partner protein fusion) were introduced by overlapping PCR. Sequences of all primers used in this work are listed in Supplementary Table 4.

**IP$_1$ production assay.** The Cisbio IP-One kit was used according to the manufacturer's instructions. HEK293 cells (ATCC CRL-1573) were seeded onto poly-L-Lysine-coated 384-well plates at 20,000 cells per well and transfected with 40 ng of DNA coding for the wild-type CysLT$_2$R or for the CysLT$_2$R mutants using the X-treme-Gene HP (Roche) agent. At 48 h post transfection, the media was removed and the cells were washed with fresh Hank's Balanced Salt Solution. Cells were either stimulated directly with a range of LTD$_4$ concentrations ($10^{-12}$–$10^{-6}$ M) prepared in IP$_1$ stimulation buffer, or sequentially stimulated with a range of antagonist concentrations ($10^{-11}$–$10^{-5}$ M), and LTD$_4$ concentrations corresponding to the EC$_{80}$ for each mutant. No LTD$_4$ degradation was observed by mass spectrometry (Supplementary Fig. 8). After equilibration for 30 min at 37 °C, the cells were lysed with IP$_1$-D2 and Ab-Crypt reagents in lysis buffer and then incubated for 1 h at RT. Fluorescence signal was recorded on a Tecan GENios Pro plate reader using an HTRF filter set ($\lambda_{ex}$ 320 nm, $\lambda_{em}$ 620 and 655 nm). Data were plotted using the three parameters EC$_{50}$/IC$_{50}$ fit in GraphPad Prism 7 (San Diego, CA) and represent the mean ± s.d. of at least two independent experiments performed in quadruplicate.

**Quantification of LTD$_4$ degradation in IP$_1$ assay.** Potential conversion of LTD$_4$ into LTC$_4$ or LTE$_4$ was checked by ultra-performance liquid chromatography coupled to mass spectrometry (UPLC/MS). HEK293 cells were seeded in a 6-well plate at a density of 300,000 cells per well. Forty-eight hours after seeding, medium was removed, and cells were washed twice with PBS. Then cells were incubated in stimulation buffer used for IP-One assays (Krebs buffer containing LiCl as an inhibitor of IP$_1$ degradation) alone or containing 10 μM LTD$_4$ or 10 μM LTD$_4$ and 10 mM L-Cysteine (used as an inhibitor of LTD$_4$ conversion) for 30 min at 37 °C. After incubation, supernatant was filtered through a 0.22 μm PVDF filter and an internal standard was added before injection on UPLC/MS (Waters UPLC system coupled with a SQ detector 2 and a PDA eλ detector, using an Acquity UPLC BEH C18 column, 2.1 mm × 50 mm, 1.7 μm spherical size). UPLC chromatograms were recorded using the following gradient: water + 0.1% TFA and acetonitrile (0 → 0.2 min, 5% acetonitrile; 0.2 → 1.5 min, 5% → 95%; 1.5 → 1.8 min, 95%; 1.8 → 2.0 min, 95% → 5%; and 2.0 → 2.5 min, 5%). Quantification was done by determining the area under the curve (AUC) ratio of the tested compound over AUC of the internal standard. 0% was determined by using results from stimulation buffer alone and 100% was determined by using results from stimulation buffer containing LTD$_4$ but without incubation over the cell monolayer. Quantification results are expressed as mean ± s.e.m. of three independent experiments.

**Cell surface expression determined by ELISA.** HEK293 cells were seeded in 24-well plates coated with poly-L-Lysine (Sigma) at 100,000 cells per well and transfected with 375 ng of plasmid coding for the wild-type or mutant CysLT$_2$R using X-treme-Gene HP (Roche). Forty-eight hours after transfection, cells were fixed with 3.7% (v/v) formaldehyde in Tris-buffered saline (TBS, 20 mM Tris-HCl, pH 7.5, and 150 mM NaCl) for 5 min at RT. Cells were washed three times with TBS and incubated for 1 h in TBS supplemented with 3% (w/v) fat-free

milk in order to block nonspecific binding sites. A mouse monoclonal anti-HA antibody coupled to HRP (Roche) was added at 1:1000 dilution in TBS-3% fat-free dry milk for 3 h at RT. Following incubation, cells were washed twice with TBS before the addition of 250 μl of 3,3′,5,5′-Tetramethylbenzidine (Sigma). Plates were incubated for 15 min at RT and the reaction was stopped by the addition of 250 μl of 2N HCl. Two hundred microliters of the yellow reaction was transferred into a 96-well plate and the absorbance was read at 450 nm on GENios Pro plate reader (Tecan). Cells transfected with the empty pcDNA3.1+ vector (mock) were used to determine background. Data were plotted using GraphPad Prism 7 and represent the mean ± s.d. of at least two independent experiments performed in quadruplicate.

**Molecular docking**. We collected 18 O- and N-derivatives of the common 3,4-dihydro-2H-1,4-benzoxazine-2-carboxylic acid scaffold from previous studies[30,32], assigned charges for the ligands at pH 7.0, and generated 3D ligand structures from their 2D representations, using Monte Carlo optimization and the MMFF-94 force field. We preprocessed each protein structure (CysLT$_1$R-pranlukast, PDB ID 6RZ4; CysLT$_2$R-11a, PDB ID 6RZ6) by adding missing residues, optimizing side-chain rotamers, and removing water molecules. Rectangular boxes enclosing ligand-binding sites of pranlukast in CysLT$_1$R and cpd 11a in CysLT$_2$R with an additional 8 Å margin were used as the sampling space for docking. Receptors were presented as smoothened grid potentials, while the docking simulations sampled ligand conformations in the internal coordinate space using biased probability Monte Carlo optimization[57] with the sampling parameter (docking effort) set to 50. We performed at least two independent docking runs for each ligand and selected binding poses with the lowest docking score. All docking simulations were done using the ICM-Pro v3.8–6 software package (MolSoft).

**MD simulations**. The initial CysLT$_2$R models for MD simulations were prepared based on the crystal structures (CysLT$_2$R-11a, PDB ID 6RZ6, for the inactive state; CysLT$_2$R-11c, PDB ID 6RZ8 for the intermediate state) using the ICM-Pro molecular modeling package (v3.8–6). First, BRIL-fusions and all hetero atoms were removed, followed by the assignments of protonation states and modeling missing side-chain residues using internal coordinate mechanics force field. Then missing loops were modeled using the loop modeling and regularization protocols available in ICM-Pro[58]. These preprocessed CysLT$_2$R models were used to prepare input files for MD simulations as previously described[59]. Briefly, the input files were generated using the CHARMM-GUI server[60]. The receptor orientation was calculated by superimposing the CysLT$_2$R structures on the CB1 receptor coordinates (PDB ID 5XRA) obtained from the OPM database[61]. The input simulation box had 157 POPC lipids, 11,908 water molecules, and 31 sodium and 46 chloride ions. The system was first energy minimized and then equilibrated for 10 ns, followed by ten independent production runs of 500 ns each using Gromacs (v.2018.1) simulation package[62]. The analysis and plotting were performed using Gromacs and matplotlib plotting packages available in Python. The MD simulations were performed on GPU enabled nodes with P100 NVIDIA cards made available by the High-Performance Computing Center at the University of Southern California.

**Ligand synthesis and characterization**. The overall ligand synthesis scheme is shown in Supplementary Fig. 1 and described in Supplementary Methods. Analytical samples were homogeneous as confirmed by TLC, and afforded spectroscopic results consistent with the assigned structures. Proton and carbon nuclear magnetic resonance spectra ($^1$H and $^{13}$C NMR) were taken on a Varian Mercury 300 spectrometer using deuterated chloroform (CDCl$_3$) and deuterated dimethyl-sulfoxide (DMSO-$d_6$) as the solvent. Fast atom bombardment mass spectra were obtained on a JEOL JMS-DX303HF spectrometer. Electrospray ionization (HRMS) mass spectra was obtained on a Thermo Fisher Scientific LTQ Orbitrap XL system. Column chromatography was carried out on silica gel (Merck Silica Gel 60, Wako gel C-200, or Fuji Silysia FL60D). Thin layer chromatography was performed on silica gel (Merck TLC or HPTLC plates, Silica Gel 60 F254).

**Reporting summary**. Further information on research design is available in the Nature Research Reporting Summary linked to this article.

## Data availability

Data supporting the findings of this manuscript are available from the corresponding authors upon reasonable request. A reporting summary for this article is available as a Supplementary Information file. The source data underlying Supplementary Figs. 5 and 6 are provided as a Source Data file. Coordinates and structure factors have been deposited in the Protein Data Bank (PDB) under the accession codes 6RZ6 (CysLT$_2$R-11a, C222$_1$ space group), 6RZ7 (CysLT$_2$R-11a, F222 space group), 6RZ8 (CysLT$_2$R-11c), and 6RZ9 (CysLT$_2$R-11b).

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

## Acknowledgements

We thank S. Ustinova, A. Awawdeh, P. Utrobin, J. Velasquez, and Yu. Kovalev for technical assistance, C. Hanson, K. Villers, and M. Chu for their help with insect and mammalian cells expression, the Structural Biology Group of the ESRF, and especially A.N. Popov for assistance with data collection. We also thank the High-Performance Computing Center at the University of Southern California for providing computing resources. This work was supported in part by the Russian Science Foundation projects 19–14–00261 (A.G., A.L., E.M., A.M., and M.S.) and 18–74–00117 (P.P.) and the GPCR consortium (V.C.). V.B. acknowledges support from the Ministry of Science and Higher Education of the Russian Federation (project 6.9909.2017/6.7). V.G. acknowledges the special agreement CEA (IBS)–HGF(FZJ) STC 5.1, the Grenoble Instruct Centre (ISBG; UMS 3518 CNRS-CEA-UJF-EMBL), the French Infrastructure for Integrated Structural Biology (FRISBI; ANR-10-INSB-05–02), and the New Generation of Drugs for Alzheimer's Disease project (GRAL; ANR-10-LABX-49–01) within the Grenoble Partnership for Structural Biology. B.S. was supported by a fellowship from EMBO (ALTF 677-2014). É.B.O. was supported by a research fellowship from the Institut de Pharmacologie de Sherbrooke and Centre d'excellence en neurosciences de l'Université de Sherbrooke. R.B. was supported by a research fellowship from the Canadian Institutes of Health Research and from the Fonds de Recherche en Santé du Québec. P.S. holds a Canada Research Chair in Neurophysiopharmacology of Chronic Pain.

## Author contributions

A.G. and A.L. optimized the constructs, developed the expression and purification procedure, expressed and purified the proteins, screened the ligands, crystallized the protein–ligand complexes, collected synchrotron data, and prepared initial draft. A.G., A.L., E.M., V.B., K.K. and A.M. collected X-ray diffraction data at synchrotron. É.B.O., R.B., J.M.L. and P.S. performed and analyzed cell signaling and cell surface assays. T.F. and T.M. provided ligands, and performed SAR analysis. E.M. and V.B. processed diffraction data. E.M., V.B. and G.W.H. performed structure determination and refinement. V.B., V.C., A.G., A.L., P.P., E.M., A.M., G.W.H. and V.K. performed project data analysis/interpretation. B.S. provided advice on construct design. A.I., A.S., and M.S. helped with construct optimization, protein expression and purification. D.R., P.P. and V.K. prepared structural models for docking, performed molecular docking, structure-activity-relationship, and structure analysis. N.P. and V.K. performed MD simulations and data analysis. A.G., A.L., V.C., P.P., V.B. and V.K. wrote the manuscript with help from other authors. V.C, V.G., A.M. and V.B. initiated the project. A.M. and V.B. organized the project implementation, were responsible for the overall project management, and cosupervised the research. V.C. supervised the overall project.

## Competing interests

T.F. and T.M. are employees of Ono Pharmaceutical. Other authors declare no competing interests.
