## [Peer Review File · Nature Communications]

Reviewers' Comments:

Reviewer #1:

Remarks to the Author:

In the manuscript "Structural Basis of Ligand Selectivity and Disease Mutations in Cysteinyl Leukotriene Receptors," A. Gusach et al. reported high-resolution crystal structures of a medically relevant receptor CysLT2R and its complexes with three novel antagonists. Combined with experimental mutagenesis and computational docking studies, ligand selectivities in CysLT2R and CysLT1R are explained. The results also provided insights into disease-related mutations. The study is thorough in the methods and analyses and is suitable for publication in Nature Communications. Some comments are listed below.

The three structures of CysLT2R complexed with 11a, 11b, and 11c show that there are significant ligand-induced structural changes in the receptor upon the ligand binding, which could even capture an intermediate conformational state as in the structure of 11c complex. Specifically, the R1 moiety of the compound series under the study seems to have a visible impact on the receptor structure. This observation raises the question of whether the receptor structure can be kept rigid in the molecular docking of 18 compounds with a variety of R1. What's the difference between the two CysLT2R structures (11a and 11c) in the R1 region? Which structure was used in the docking study?

Page 16, line 336. A 70-80 degree of data will give a nearly complete data set. 90 degree is needed for a complete data set in space group I4.

Page 39, Sup Fig 1. The namings of 11a,b,c are not consistent between the sketch and the figure legend.

Page 40, Sup Fig 2. For the completeness, data for 11b should be included as well.

Page 41. Sup Fig 3. From the patterns in 3b, it seems that the b-axis (142A) could be halved. Wondering whether the F222 cell could be reduced to a smaller unit cell. The projections on a-axis and b-axis are identical for space group I4. A projection along with the a+b direction could show more packing patterns.

Reviewer #2:

Remarks to the Author:

The research article by A.Gusach et al, reports four X-ray structures of CysLT2R in complex with three novel dual CysLT1R/CysLT2R antagonists. Thermostabilised apocytochrome b562RIL fusion protein and thermostabilising mutations were used to obtain high-quality diffracting crystals of the CysLT2R ligand-bound conformation. The paper describes the canonical seven-transmembrane helical bundle architecture, binding site and conserved motif/residues known for their role in stabilising active and inactive receptor conformations or for receptor signalling. The manuscript is clear and well written and the structural analysis sheds light on molecular mechanism of disease-associated single nucleotide variants related to asthma (M2015.38V) and for one oncogenic mutations (L1293.43Q). Based on structural analysis, the authors provide a rationale for the effects of specific SNVs on receptor function. This study is a nice proof of concept for the use of structural characterisation for predicting disease associations and effects of natural missense variants on drug efficacy.

1- Co-crystallised ligands are reported to be antagonist although 11a- and 11b-bound conformations display residues and motifs corresponding to active state of the receptor. In case of 11c, the overall structure is even described as an active like conformation, which is rather

surprising for antagonist-bound receptor structure. Usually antagonist-bound conformations are distinct from agonist-bound conformations, although differences can be very subtle. Can the authors comment on such results?

2- Page 9 « Substitutions of L1985.35 with alanine or M2015.38 with alanine or leucine result in non-responsive mutants that bind LTD4 but fail to stimulate IP1 production. The atopic asthma-associated variant M2015.38V does not completely abolish signalling but significantly decreases LTD4 potency and efficacy in IP1 assay ». M2015.38V signalling is reduced but still functional, and this mutant responds to LTD4. The authors may want to clarify this sentence.

3- It might be nice have a 2D plot of all three ligand, 11, a, b, and c. I could not find 11b.

4- The receptor activity cannot be reverse by compounds 11a or 11c, according to Supp Fig. 6d, they do even stimulate IP production, Is this correct? Is there any possible explanation for such pharmacological effect?

5- Statistical analysis is missing for Supp Fig. 6d and LTD4 stimulation does not look very convincing.

Reviewer #3:

Remarks to the Author:

The study by Gusach et al. is a valuable and insightful addition to the literature on class-A GPCRs, providing useful biomedical information on the effect of disease-related mutations and selective antagonists on cysteinyl leukotriene receptors (CysLTR). The work is very thorough, combining new crystal structures of CysLT2R, compound structure-activity analyses, computational studies and the mapping of genetic data. The detection of an active-like structure in the ensemble is a particularly interesting new finding in my view, as is the structural mapping of single nucleotide variants (SNV).

The computational investigations have been conducted with the appropriate rigour, repeating the molecular dynamics simulations five times and docking a range of compounds. In combination with the rest of the findings, I think that no new computational experiments are needed to underpin the main messages of the manuscript.

Regarding the mapping of disease-related SNVs to the structures, I agree that the mutation L3.43Q is likely to disrupt the hydrophobic layer below the sodium binding site. The authors argue that this would facilitate water and sodium passage and thereby favour the active state, however the cited reference (36) does not relate to sodium passage. Activation-related sodium passage across this hydrophobic region was shown in simulations by Vickery et al., *Structure* 2018 (doi: 10.1016/j.str.2017.11.013).

I would be interested in a somewhat more elaborate discussion of the relevance of the active-like state for elucidating activation pathways and comparison to other intermediate states in the literature, either from previous structure analyses or simulations. In particular, one simulation (Fig S7b, bottom) appears to fall back from the active-like into the inactive state - are there any changes in key microswitch conformations that accompany this transition and if so, is there a sequence of events?

Minor:

The circles representing the position of the crystal structures on the 2D-graph in Fig S7 are a bit hard to see, perhaps they can be replaced by a different colour or larger symbol.

Finally, it would be interesting to know if the simulations were performed on the thermostabilised mutants.

We would like to thank Reviewers for their helpful comments and constructive critique, which we fully addressed below. The manuscript was revised accordingly, and all changes are highlighted in yellow. Our point-by-point responses to the Reviewers' comments are shown below in bold:

Reviewer #1 (Remarks to the Author):

In the manuscript "Structural Basis of Ligand Selectivity and Disease Mutations in Cysteinyl Leukotriene Receptors," A. Gusach et al. reported high-resolution crystal structures of a medically relevant receptor CysLT₂R and its complexes with three novel antagonists. Combined with experimental mutagenesis and computational docking studies, ligand selectivities in CysLT₂R and CysLT₁R are explained. The results also provided insights into disease-related mutations. The study is thorough in the methods and analyses and is suitable for publication in Nature Communications. Some comments are listed below.

The three structures of CysLT₂R complexed with 11a, 11b, and 11c show that there are significant ligand-induced structural changes in the receptor upon the ligand binding, which could even capture an intermediate conformational state as in the structure of 11c complex. Specifically, the R1 moiety of the compound series under the study seems to have a visible impact on the receptor structure. This observation raises the question of whether the receptor structure can be kept rigid in the molecular docking of 18 compounds with a variety of R1. What's the difference between the two CysLT₂R structures (11a and 11c) in the R1 region? Which structure was used in the docking study?

Although the overall variations between the structure of the 11c complex and two other structures are significant, the changes in the conformations of the ligand-binding pocket residues are relatively minor (all-atom RMSD < 0.6 Å). The largest deviation among the residues in the vicinity of the binding pocket is only 0.7 Å for Tyr127 OH moiety (notably, this residue does not engage in direct contacts with any of the docked ligand). Therefore, for docking study we used the highest resolution structure of the CysLT₂R-11a complex (PDB ID 6RZ6).

Page 16, line 336. A 70-80 degree of data will give a nearly complete data set. 90 degree is needed for a complete data set in space group I4.

We thank the reviewer for catching this inaccuracy. The sentence has been corrected as follows: "In case of CysLT₂R-11c, four partial (70-80°) datasets with oscillation 0.2° and three partial 20° datasets with oscillation 0.1° were collected and combined to obtain a complete final dataset."

Page 39, Sup Fig 1. The namings of 11a,b,c are not consistent between the sketch and the figure legend.

We apologize for this oversight. The mistake has been corrected so all the namings are now consistent.

Page 40, Sup Fig 2. For the completeness, data for 11b should be included as well.

We have updated Sup Fig 2 to include data for all three ligands.

Page 41. Sup Fig 3. From the patterns in 3b, it seems that the b-axis (142A) could be halved. Wondering whether the F222 cell could be reduced to a smaller unit cell. The projections on a-axis and b-axis are identical for space group I4. A projection along with the a+b direction could show more packing patterns.

We have revised Sup Fig 3 to label the unit cell axes and to include a projection along the a+b direction for the I4 cell. From the left image of panel (b) it is clear that the b-axis could not be halved. We have also checked the spacegroup by the program Zanuda confirming correct assignment.

Reviewer #2 (Remarks to the Author):

The research article by A.Gusach et al, reports four X-ray structures of CysLT₂R in complex with three novel dual CysLT₁R/CysLT₂R antagonists. Thermostabilised apocytochrome b562RIL fusion protein and thermostabilising mutations were used to obtain high-quality diffracting crystals of the CysLT₂R ligand-bound conformation. The paper describes the canonical seven-transmembrane helical bundle architecture, binding site and conserved motif/residues known for their role in stabilising active and inactive receptor conformations or for receptor signalling. The manuscript is clear and well written and the structural analysis sheds light on molecular mechanism of disease-associated single nucleotide variants related to asthma (M2015.38V) and for one oncogenic mutations (L1293.43Q). Based on structural analysis, the authors provide a rationale for the effects of specific SNVs on receptor function. This study is a nice proof of concept for the use of structural characterisation for predicting disease associations and effects of natural missense variants on drug efficacy.

1- Co-crystallised ligands are reported to be antagonist although 11a- and 11b-bound conformations display residues and motifs corresponding to active state of the receptor. In case of 11c, the overall structure is even described as an active like conformation, which is rather surprising for antagonist-bound receptor structure. Usually antagonist-bound conformations are distinct from agonist-bound conformations, although differences can be very subtle. Can the authors comment on such results?

We thank the reviewer for this comment. This point indeed needs some clarification. We included additional explanations for the conformational state of CysLT₂R in the text (section “Overall architecture of CysLT₂R”). Specifically, we emphasized that 11a and 11b-bound conformations are fully inactive. Like in other GPCR-antagonist complexes from the delta branch (including CysLT₁R), the P^{5.50}-I^{3.40}-F^{6.44} motif in CysLT₂R was found in a distinct conformation reminiscent of an active-like state. However, analysis of other delta-branch structures shows that this conformation is not a hallmark of an active state, but rather a result of the replacement of the “toggle switch” W6.48 with Y6.48, which stabilizes this P-I-F conformation even in the inactive antagonist-bound structures.

Regarding the “intermediate” state of the 11c complex, it does have some features of an active-like state, specifically, an outward movement of TM6, accompanied by a switch of Y^{5.58} that supports the TM6 conformation. However, as shown in our MD simulations, this “active-like” state in CysLT₂R is highly dynamic and transient. This state was likely captured and stabilized by the crystal lattice, enabling us to determine its structure.

2- Page 9 « Substitutions of L1985.35 with alanine or M2015.38 with alanine or leucine result in non-responsive mutants that bind LTD4 but fail to stimulate IP1 production. The atopic asthma-associated

variant M2015.38V does not completely abolish signalling but significantly decreases LTD₄ potency and efficacy in IP₁ assay ». M2015.38V signalling is reduced but still functional, and this mutant responds to LTD₄. The authors may want to clarify this sentence.

To clarify, we rephrased the second part of this sentence it as follows:

In contrast to the alanine or leucine substitution, the atopic asthma-associated variant M201^{5.38}V still responds to LTD₄ stimulation. However, this mutation significantly decreases LTD₄ potency and efficacy to induce IP₁ accumulation when compared with the wild type CysLT₂R (Table 1).

3- It might be nice have a 2D plot of all three ligand, 11, a, b, and c. I could not find 11b.

We agree with the reviewer and have included ligand 11b in the 2D plot presented in Fig. 2c.

4- The receptor activity cannot be reverse by compounds 11a or 11c, according to Supp Fig. 6d, they do even stimulate IP production, Is this correct? Is there any possible explanation for such pharmacological effect?

As correctly stated by the reviewer, we show in Supp. Fig. 6d that the compounds 11a and 11c do not reverse the constitutive activity of the L129Q mutant. However, these compounds do not significantly increase IP₁ production at this receptor. We agree that the slight sigmoidal curves shown in this figure for 11a and 11c compounds might lead to confusion. We have performed statistical analysis on these data and confirmed that compounds 11a and 11c neither reverse L129Q constitutive activity, nor induce IP₁ production at this mutant.

In order to clarify this for the reader, we decided to replace both Suppl. Fig. 6c and 6d by a new bar graph (new Supplemental Fig. 6c). Please note that, for this figure, we pooled together all of our data with the L129Q mutant, rather than solely presenting the data associated with the full concentration-response curves. We also chose to present the normalized data, rather than as nanomolar concentrations of IP₁.

5- Statistical analysis is missing for Supp Fig. 6d and LTD₄ stimulation does not look very convincing.

We have applied statistical analysis for the new Supplemental Fig. 6c.

Reviewer #3 (Remarks to the Author):

The study by Gusach et al. is a valuable and insightful addition to the literature on class-A GPCRs, providing useful biomedical information on the effect of disease-related mutations and selective antagonists on cysteinyl leukotriene receptors (CysLTR). The work is very thorough, combining new crystal structures of CysLT₂R, compound structure-activity analyses, computational studies and the mapping of genetic data. The detection of an active-like structure in the ensemble is a particularly interesting new finding in my view, as is the structural mapping of single nucleotide variants (SNV).

The computational investigations have been conducted with the appropriate rigour, repeating the molecular dynamics simulations five times and docking a range of compounds. In combination with the rest of the findings, I think that no new computational experiments are needed to underpin the main messages of the manuscript.

Regarding the mapping of disease-related SNVs to the structures, I agree that the mutation L3.43Q is likely to disrupt the hydrophobic layer below the sodium binding site. The authors argue that this would facilitate water and sodium passage and thereby favour the active state, however the cited reference (36) does not relate to sodium passage. Activation-related sodium passage across this hydrophobic region was shown in simulations by Vickery et al., Structure 2018 (doi: 10.1016/j.str.2017.11.013).

We have added a reference to Vickery et al., 2018 as suggested.

I would be interested in a somewhat more elaborate discussion of the relevance of the active-like state for elucidating activation pathways and comparison to other intermediate states in the literature, either from previous structure analyses or simulations. In particular, one simulation (Fig S7b, bottom) appears to fall back from the active-like into the inactive state - are there any changes in key microswitch conformations that accompany this transition and if so, is there a sequence of events?

In response to this question, and a similar question from Reviewer 2 above, we added a brief explanation of the observed conformational states (Overall architecture of CysLT2R section), putting them in the context of other delta-branch GPCR structures. Regarding the specific question about the MD results shown in Fig S7b, we checked that while TM6 sometimes reverts from an intermediate to a fully inactive state, the P-I-F does not switch its conformation, and the sodium pocket maintains inactive conformation with Na⁺ ion bound throughout the simulation.

Minor:

The circles representing the position of the crystal structures on the 2D-graph in Fig S7 are a bit hard to see, perhaps they can be replaced by a different colour or larger symbol.

We thank the review for this suggestion. We have changed the size of the symbols in Fig S7c, so that the circles are now easier to see.

Finally, it would be interesting to know if the simulations were performed on the thermostabilised mutants.

Yes, the MD simulations were performed on the thermostabilised mutants. We have clarified this point in the Methods.

Reviewers' Comments:

Reviewer #1:

Remarks to the Author:

The revision has addressed all my questions.

Reviewer #2:

Remarks to the Author:

All concerns have been properly addressed, and this research article is suitable for publication.

Reviewer #3:

Remarks to the Author:

I am happy with the revision; all my points have been addressed. This is a very thorough and insightful paper.